# Controllable Image Captioning with Feature Refinement and Multilayer Fusion

**Sen Du** [1], **Hong Zhu** [1,*], **Yujia Zhang** [1], **Dong Wang** [1], **Jing Shi** [1], **Nan Xing** [1], **Guangfeng Lin** [2] and **Huiyu Zhou** [3]

1 School of Automation and Information Engineering, Xi'an University of Technology, Xi'an 710048, China
2 School of Printing, Packaging and Digital Media, Xi'an University of Technology, Xi'an 710054, China
3 School of Computing and Mathematical Sciences, University of Leicester, University Road, Leicester LE1 7RH, UK
* Correspondence: zhuhong@xaut.edu.cn

**Abstract:** Image captioning is the task of automatically generating a description of an image. Traditional image captioning models tend to generate a sentence describing the most conspicuous objects, but fail to describe a desired region or object as human. In order to generate sentences based on a given target, understanding the relationships between particular objects and describing them accurately is central to this task. In detail, information-augmented embedding is used to add prior information to each object, and a new Multi-Relational Weighted Graph Convolutional Network (MR-WGCN) is designed for fusing the information of adjacent objects. Then, a dynamic attention decoder module selectively focuses on particular objects or semantic contents. Finally, the model is optimized by similarity loss. The experiment on MSCOCO Entities demonstrates that IANR obtains, to date, the best published CIDEr performance of 124.52% on the Karpathy test split. Extensive experiments and ablations on both the MSCOCO Entities and the Flickr30k Entities demonstrate the effectiveness of each module. Meanwhile, IANR achieves better accuracy and controllability than the state-of-the-art models under the widely used evaluation metric.

**Keywords:** controllable image captioning; information-augmented embedding; MR-WGCN; similarity loss

## 1. Introduction

Image captioning is a complex task of automatically producing natural language sentences to describe the content of a given image. It requires not only understanding the relationship between each object, but also generating sentences to describe the most conspicuous objects.

In the early stages, the image captioning task is based on templates [1,2] or retrieval [3,4]. In recent years, with the rapid development of deep learning [5,6], most current image captioning methods have been based on deep learning and adopted an encoder–decoder structure, in which the encoder extracts features to represent the content of the image. The decoder uses the extracted features to generate a description. The latest caption model even achieved better performance than humans in all accuracy-based metrics. However, these established models tend to generate sentences by describing the most conspicuous objects, but fail to describe a desired region or object as human. It is essential for practical applications. For example, when assisting visually impaired people to walk, the generated caption should describe what is on the road or the color of traffic lights. Meanwhile, many studies have indicated that traditional models tend to produce generic descriptions to capture frequent descriptive patterns, but fail to describe particular objects. To endow captioning models with controllability, several models introduce extra control signals to generate captions, called controllable image captions (CIC).

The CIC model can easily generate diverse captions for the same image by feeding different control signals. One type of CIC [7–9] focuses on controlling describing styles,

such as factual, sadly, and happy; the other type aims to control the content, such as region [10], object [11,12], part-of-speech tags [13,14], and length level [15].

To produce controllable object image captioning, Marcella et al. [11] proposed a model to control the content and the order of the image caption explicitly grounded on a sequence of image regions. Chen et al. [16] proposed a control signal that represents a targeted activity as a verb and some entities involved in this activity as semantic roles. Many researchers [12,14,15] work in this direction.

In current models, the accuracy of the generated sentences depends on the accuracy of the understanding of the object role. However, the object feature obtained by the detection model lacks prior information. For example, given the features of a man and a baseball, it would be difficult to infer their relationship and the concept of a player. Besides, most of the models adopt cross-entropy loss, which leaves a lack of diversity in the generated sentences.

Based on the above problems, this paper introduces a framework based on the information-augmented and node-relation estimation network (IANR) to improve the performance of controllable image captioning. This method is an encoder–decoder structure. The information-augmented graph encoder consists of an information-augmented embedding module and a multi-relational weighted graph convolution network (MR-WGCN). The information-augmented embedding module is designed to add prior information for objects and relationships. The MR-WGCN emulates the message passing from one node to others in different ways. In terms of the decoder, this paper designs a model that dynamically pays attention to control signals or features with prior information. To further increase the diversity of descriptions, an additional similarity loss is added to the traditional cross-entropy loss.

The main contributions are summarized as follows:

- The proposed information-augmented embedding module adds prior information for each object and relation node.
- A Multi-Relational Weighted Convolution Graph (MR-WGCN) is proposed to aggregate messages from related nodes in different ways.
- A dynamic attention decoder is designed to fuse the result of control signals or node features with prior information, which can address the need to generate sentences that satisfy the control signal.
- The designed novel similarity loss cooperate with traditional cross-entropy loss to utilize information effectively for generating diverse captions.
- This paper performs an extensive comparative study on two commonly used datasets, et al., MSCOCO Entities and Flickr30k Entities, to evaluate designed IANR. The experimental results show that the proposed method achieves significantly higher accuracy and diversity in all evaluation metrics than the baseline method, ASG2Caption. In addition, IANR achieves state-of-the-art controllability and accuracy on two datasets.

The rest of the paper is structured as follows. First, related work is briefly introduced and discussed in Section 2. Section 3 introduces the proposed method for controllable image captioning (CIC). Section 4 shows the experimental evaluation of the proposed method and other methods. Finally, Section 5 discusses the conclusion and future research directions.

## 2. Related Work

At present, most image captioning models have achieved significant improvement based on the encoder–decoder and reinforcement learning. Inspired by neural language translation, the encoder–decoder structure learns image content with an encoder and transforms the image content into sentences with a decoder. The NIC [17] exploits the convolution neural network to obtain a fixed-length vector representing the content of the whole image and recurrent neural networks to generate words sequentially. Traditional image captioning methods are trained by maximizing the likelihood of ground truth captions, which cannot optimize quality metrics, such as CIDEr. Self-critical Refs. [18–20] optimized non-differentiable metrics using reinforcement learning. To reduce the impact of redundant regions in the image, Refs. [21,22] encodes the features of detected object

regions [23], and then ground words with relevant image regions dynamically in generation. Except for the detected region, some researchers regard the sentences as the relationships of the objects in the image. The Refs. [24–26] adopted the scene graph [27] to utilize the detected objects and their relationships. The ASG2Caption [12] proposed an abstract scene graph (ASG) instead of the detected scene graph to generate the desired caption. This work proposes a novel module called the information-augmented graph encoder, which is composed of an information-augmented embedding module and a multi-relational weighted graph encoder to incorporate a priori information into objects or relation nodes, improving the accuracy and diversity of the generated sentences.

*Controllable Image Captioning*

Controllable image captioning is a more challenging task that aims to generate sentences according to extra control signals, such as style and semantic. The target of style control research [7–9,28,29] is to restrain emotions or linguistic styles, such as factual, sad, happy, or humorous. etc.Most of them [28,30–32] train on datasets with stylized labels. A few studies [7,33] use a monolingual stylized language corpus without paired images to disentangle style from factual items.

The target of semantic control aims to control the described contents or structures in the image, such as region [10], object [11,12,34,35], part-of-speech tags [13,14], and length level [15]. DenseCap [10] detects and describes diverse regions in the image. CGO [34] combines two LSTMs in opposite directions for generating image captions with desired objects. SCT [11] controls the described objects and the order of the generated sentences. ASG2Caption [12] proposes an abstract scene graph to control the described objects and relationships. Sub-GC [36] describes sub-graphs of image scene graphs. POS [13] uses the Part-of-Speech tag sequence to guide caption generation. MTTSNet [14] generates sentences with the assistance of POS information for each relationship between object combinations in a scene graph. LaBERT [15] uses a length signal to control and describe the image, either roughly or in detail. In addition to this, there are other semantic control methods, such as DUDA [37] that describes semantic differences between two images. SCAN [38] introduces a signal controlling the sentence quality, sentence length, and number of nouns.

All the above work mainly concentrates on the control process. They usually adopt region features as one of the inputs, ignoring the prior information about the objects and their relationships. This paper not only proposes an information-augmented graph encoder to add prior information to each node, but also proposes an improved dynamic attention decoder to selectively focus on the control signals or node features. Finally, the proposed similarity loss facilitates IANR to learn more diverse information.

## 3. Proposed Model

Given an image $I$, the goal of IANR is to generate a fluent caption $y = \{y_1, \cdots, y_T\}$ based on the control signal of the Abstract Scene Graph (ASG) [12]. The ASG reflects the user's intention through nodes and their relationships. Humans can describe the given image through multiple sentences. Meanwhile, the image has multiple ASGs. According to different ASGs, IANR can generate different sentences. The structure of IANR is shown in Figure 1. The structure includes an information-augmented graph encoder and a dynamic attention decoder. Section 3.1 describes how the proposed information-augmented graph encoder adds prior information and uses the proposed MR-WGCN to enhance features with surrounding node information. Section 3.2 describes how to generate sentences and update all node information. Finally, we train the IANR through cross-entropy loss and the similarity loss.

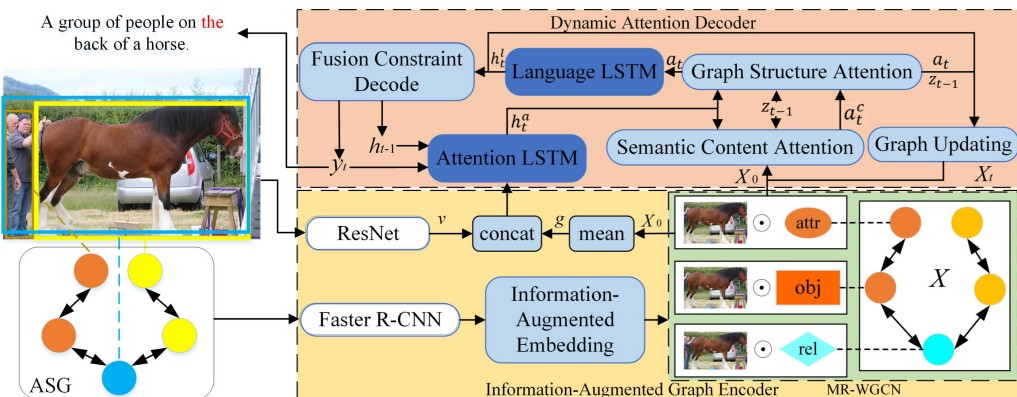

**Figure 1.** The IANR includes an Information-Augmented Graph Encoder and a Dynamic Attention Decoder. Given an image *I* and a control signal ASG $\boldsymbol{G} = (X, E)$, *X* and *E* are the sets of nodes and edges. The information-augmented embedding adds a priori information to all nodes. The proposed MR-WGCN enhances node features through surrounding information. Then the Dynamic Attention Decoder incorporates the result of Semantic Content Attention and Graph Structure Attention to select node information. Finally, this paper generates sentences through language LSTM and the proposed Fusion Constraint Decode module. After generating a word, we updated the node feature of graph $X_{t-1}$ to $X_t$.

### 3.1. Information-Augmented Graph Encoder

The encoder was proposed to encode ASG as a set of node features $\boldsymbol{X} = \{x_1, \cdots, x_N\}$, where *N* is the number of nodes in ASG. The ASG for image *I* was denoted as $\boldsymbol{G} = (\boldsymbol{X}, \boldsymbol{E})$, where *X* and *E* are the sets of nodes and edges, respectively. The types of nodes *X* are object node *o*, attribute node *a*, and relationship node *r*. The six types of edges *E* are bidirectional connections between the subject node *o* and relationship node *r*, object node *o* and attribute node *a*, and relationship node *r* to object *o*, respectively. The feature of nodes in ASG was extracted from the grounded box in the image. The box of the relationship node was the union bounding box of the two involved objects.

ASG contains information about each node region, while the region features extracted from the detection network do not contain prior knowledge. For example, given the features of a man and a baseball, it would be difficult to infer their relationship and the concept of a player. To overcome this problem, the proposed information-augmented graph encoder consists of information-augmented embedding and a multi-relational weighted convolution graph to add prior information.

**Information-Augmented Embedding.** In this encoder, the memory-augmented attention operator (MA) [39] adds prior information for each node. The operator is defined as:

$$\tilde{\boldsymbol{X}} = \text{softmax}\left(\frac{W_q \boldsymbol{X} \boldsymbol{K}^T}{\sqrt{d}}\right) \boldsymbol{V}$$
$$\boldsymbol{K} = [\boldsymbol{W}_k \boldsymbol{X}; \boldsymbol{M}_k]$$
$$\boldsymbol{V} = [\boldsymbol{W}_v \boldsymbol{X}; \boldsymbol{M}_v] \tag{1}$$

where $\boldsymbol{W}_q, \boldsymbol{W}_k, \boldsymbol{W}_v \in \mathbb{R}^{C_1 \times C_2}$ are embedding matrices, $\boldsymbol{M}_k, \boldsymbol{M}_v \in \mathbb{R}^{C_2}$ are learnable matrices for a priori information, *d* is a scaling factor, and $[\cdot; \cdot]$ indicates concatenation.

The MA adds prior information to nodes and enables higher attention to the focal node based on the interrelationship of each node. However, is it possible that a node does not contain any relationships or priors? The last step of memory-augmented attention is a weighted summation of node features, which may lead to some misinterpretation. Therefore, the relational discriminator was designed to remove or modify some incorrect and unnecessary features.

In this discriminator, the sentinel value is first calculated:

$$S = \sigma\left(W_{i,g}\bar{X} + W_{h,g}X + B_s\right),\tag{2}$$

where $W_{i,g} \in \mathbb{R}^{C_2 \times C_3}$, $W_{h,g} \in \mathbb{R}^{C_1 \times C_3}$ and $B_s \in \mathbb{R}^{C_3}$ are learnable weights. $\sigma$ is the sigmoid logistic function. For the sentinel value $S \in \mathbb{R}^{N \times C_3}$, a higher $S_{i,j}$ means that the feature needs to be saved.

After that, we mixed the original node feature $X$ with the memory-augmented feature $\bar{X}$ as follows:

$$X_v = \sigma(W_1\bar{X} + W_2X + B_v),\tag{3}$$

where $W_1 \in \mathbb{R}^{C_2 \times C_3}$, $W_2 \in \mathbb{R}^{C_1 \times C_3}$ and $B \in \mathbb{R}^{C_3}$ are learnable weights.

Finally, the proposed information-augmented embedding module produces the feature $\bar{X}_v$ by rescaling $X_v$ with the sentinel value $S$:

$$\bar{X}_v = X_v \odot S\tag{4}$$

where $\odot$ represents element-wise multiplication. The node features are $\bar{X}_v \in \mathbb{R}^{N \times C_3}$. The effectiveness and accuracy of the region feature with prior information were improved by using the proposed relational discriminator structure.

**Multi-Relational Weighted Convolution Graph.** The types of nodes in ASG are object, attribute, and relationship. Since the types of nodes cannot be distinguished by their visual appearance alone, the node features were enhanced by the type embedding as follows[12]:

$$x_{rn,i}^{(0)} = \begin{cases} \bar{x}_{v,i} \odot W_{r,0}, & \text{if } i \in o; \\ \bar{x}_{v,i} \odot (W_{r,1} + \text{pos}[i]), & \text{if } i \in a; \\ \bar{x}_{v,i} \odot W_{r,2}, & \text{if } i \in r. \end{cases}\tag{5}$$

where $\bar{x}_{v,i}$ is one node in $\bar{X}_v$. $W_{r,n} \in \mathbb{R}^{C_3}(n = 0, 1, 2)$ is the embedding for three types of nodes, and $\text{pos}[i]$ is a positional embedding matrix to distinguish the different attribute nodes connected to the same object.

With the above formula, attribute information was added to the node feature. Furthermore, there were six types of edges and three types of nodes in ASG. Since nodes and edges are of different types, how does the message pass from one type of node to another along different edges? Therefore, the designed multi-relational weighted convolution graph (MR-WGCN) extends the MR-GCN [40] with different weights of edges as follows:

$$x_{rn,i}^{(l+1)} = \sigma\left(W_s^{(l)}x_{rn,i}^{(l)} + \sum_{\tilde{r} \in \mathcal{R}} \sum_{j \in \mathcal{N}_i^{\tilde{r}}} \frac{1}{w_{i,j}^{(l)}} W_{\tilde{r}}^{(l)} x_{rn,j}^{(l)}\right),\tag{6}$$

where $\mathcal{N}_i^{\tilde{r}}$ denotes the neighbours of $i$-th node under the edge $\tilde{r} \in \mathcal{R}$, $w_{i,j}^{(l)}$ is the weight of the edge from $x_{rn,i}^{(l)}$ to $x_{rn,j}^{(l)}$. $\sigma$ is the ReLU function, and $W_s^{(l)}, W_{\tilde{r}}^{(l)} \in \mathbb{R}^{C_3 \times C_3}$ are learnable matrices of $l$-th for self-loop. The formula of $w_{i,j}^{(l)}$ is:

$$w_{i,j}^{(l)} = \text{sigmoid}\left(W_i x_{rn,i}^{(l)}\left(W_j x_{rn,j}^{(l)}\right)^T\right)\tag{7}$$

Exploiting this layer brings context information from neighborhood nodes to the center node. Stacking multiple MR-GCN layers enabled us to obtain contextual context information. After that, calculating the average of $X_{rn} = \{x_{rn,0}, \cdots, x_{rn,N}\}$ as the global graph representation $g = \frac{1}{N}\sum_{i=0}^{N} X_{rn,i}$.

### 3.2. Dynamic Attention Decoder

The language decoder employs a two-layer LSTM structure [21], Semantic Content Attention, Graph Structure attention, and a Fusion Constraint Decoder. The two-layer LSTM includes an attention $LSTM_a$ and a language $LSTM_l$. The attention $LSTM_a$ computes the $h_t^a$ as follows:

$$h_t^a = LSTM_a\left(\left[W_v[g; v]; w_{t-1}; h_{t-1}^l\right], h_{t-1}^a; \theta^a\right),\tag{8}$$

where $v \in \mathbb{R}^{C_g}$ is a global image representation extracted from ResNet152, $w_{t-1}$ is the previous word embedding, $h_{t-1}^l$ is the previous hidden state from language $LSTM_a$, $[\cdot; \cdot]$ indicates concatenation, and $W_v \in \mathbb{R}^{(C_3+C_g) \times C_3}$ is a learnable matrix for dimension reduction.

Considering that ASG is a graph-based structure, there are two types of attention based on semantic content and graph structure.

**Semantic Content Attention.** Semantic Content Attention mainly takes the semantic content into account. In the following formula, $x_{t=0,i}$ is initialized to $x_{rn,i}$. Then we adjust feature $h_t^a$ through Formula (9), which is similar to the shortcut connection. Finally, the importance of each node is normalized by a softmax function.

$$\bar{h}_t^a = W_{xc}h_t^a \odot h_t^a + h_t^a \tag{9}$$

$$\bar{\alpha}_{t,i}^c = W_c \tanh\left(\bar{h}_t^a + x_{t,i}\right) \tag{10}$$

$$\alpha_t^c = \text{softmax}(\bar{\alpha}_t^c) \tag{11}$$

where $W_{xc} \in \mathbb{R}^{C_3 \times C_3}$ and $W_c \in \mathbb{R}^{C_3 \times 1}$ are learnable parameters in semantic content attention.

**Graph Structure Attention.** The Graph Structure Attention takes into account the graph structure of each node. ASG reflects the user's intended order. According to the structure of ASG, if the current node is an object node, the next node to be described will be a relation node or an attribute node close to the object node. The next node still has a lower probability of being another object node that has a common relationship with the current object node. Thus, there are three types of attention transfer: (1). Stay at the same node to describe the object with several words; (2). move to the next node to describe the relation or attribute; and (3). move to another object node that is related to the same relation node. Hence, $\alpha_{t,i}^f = \left(M_f\right)^i \alpha_{t-1}$ represents the attention transfer from the original node.

To calculate the probability of each attention transfer, $h_t^a$ was combined with the previous attention feature $z_{t-1}$ as the state feature $s_t$.

$$s_t = [h_t^a, z_{t-1}] \tag{12}$$

Then, we calculated the weight of three attention transfers through the state feature $s_t$ as follows:

$$w_g = \text{softmax}\left(ReLU\left(W_{gf}\left(W_{gl}s_t * W_{gr}s_t\right)\right)\right) \tag{13}$$

$$\alpha_t^f = \sum_{k=0}^{2} w_{g,k}\alpha_{t,k}^f \tag{14}$$

where $W_{gl}, W_{gr} \in \mathbb{R}^{2C_3 \times C_3}$ and $W_{gf} \in \mathbb{R}^{C_3 \times 3}$ are learnable parameters, and $w_g$ indicates the weight of each attention transfer.

The final step is to fuse the results of semantic content attention and graph structure attention, as follows:

$$g_t = \text{sigmoid}\left(ReLU\left(W_f(W_l s_t * W_r s_t)\right)\right) \tag{15}$$

$$\alpha_t = g_t\alpha_t^c + (1 - g_t)\alpha_t^f, \tag{16}$$

where $W_l, W_r \in \mathbb{R}^{2C_3 \times C_3}$ and $W_f \in \mathbb{R}^{C_3 \times 1}$ are learnable parameters, $g_t$ is a sentinel value to decide whether to pay more attention to the result of semantic content attention or graph structure attention.

Based on the nodes that should be focused on to obtain the current attention feature $z_t = \sum_{i=1}^{|\mathcal{N}|} \alpha_{t,i} x_{t,i}$ was used to generate the next word.

**Fusion Constraint Decoder.** The fusion constraint decoder generates the next word with the current attention feature $z_t$, the hidden state of attention $\text{LSTM}_a$, $h_t^a$, and the previous word $h_{t-1}^l$. Firstly, the hidden feature $h_t^l$ is generated by the language $\text{LSTM}_l$ of the two-layer LSTM structure [21].

$$h_t^l = \text{LSTM}_l\left([z_t; h_t^a], h_{t-1}^l; \theta^l\right) \tag{17}$$

In the standard method, the next word was generated as follows:

$$p(y_t \mid y < t) = \text{softmax}\left(W_p h_t^l + b_p\right) \tag{18}$$

To generate more accurate sentences, the following formula was used instead of the standard method.

$$w_{t1} = W_{p,1} h_t^l \tag{19}$$

$$w_{t2} = W_{p,2} h_t^l \tag{20}$$

$$p(y_t \mid y_{<t}) = \text{softmax}(w_{t1} * w_{t2}) \tag{21}$$

where $W_{p,1}, W_{p,2} \in \mathbb{R}^{C_3 \times C_w}$, $C_w$ is the number of total words. Formula (19) generates two word probabilities $w_{t1}, w_{t2}$ through $h_t^l$. The output word is the one with the highest probability according to $w_{t1}$ and $w_{t2}$. This design enabled us to obtain a more accurate result.

Meanwhile, in order for the two fully connected layers to learn different emphases of $h_t$, we generated different vectors through the similarity loss.

$$L_{\text{cosine},t} = \max(0, \cos(w_{t1}, w_{t2}) - thr_2) \tag{22}$$

A smaller $cos()$ indicates that they are less similar. $thr_2$ is a similarity threshold. If $\cos(w_{t1}, w_{t2})$ is less than $thr_2$, the two vectors are sufficiently dissimilar to be excluded from the loss. Through this loss, the probability of non-correct words is as orthogonal as possible.

Hence, using the standard cross-entropy loss and similarity loss to train IANR:

$$L = -\log \sum_{t=1}^{T} p(y_t \mid y_{<t}, \mathcal{G}, \mathcal{I}) + \gamma \sum_{t=1}^{T} L_{cosine,t} \tag{23}$$

where $\gamma$ is a weight to balance the cross-entropy loss and similarity loss.

**Graph Updating.** A sentence contains not only visual words, but also some non-visual words, such as "a", "the" and "some". When having non-visual words, the generated words do not express the accessed graph nodes, and thus the graph should not be updated. Therefore, a sentinel gate is proposed to dynamically adjust the attention weight through the output of language $\text{LSTM}_l$, $h_t^l$, and currently accessed node vector $z_t$ as follows:

$$u_t = W_u\left(W_{uh} h_t^l * W_{uz} z_t\right), \tag{24}$$

where $W_{uh}, W_{uz}, W_u \in \mathbb{R}^{C_3 \times C_3}$. $u_t$ is a vector to indicate whether or not the generated word expresses the meaning of the accessed node.

As with NMT [6], the node update by using the add operation after an erase operation is as follows:

$$e_{t,i} = \text{sigmoid}(W_e(W_{eu}u_t * W_{ex}x_{t,i})) \tag{25}$$

$$a_{t,i} = W_a(W_{au}u_t * W_{ax}x_{t,i}) \tag{26}$$

$$x_{t+1,i} = x_{t,i} * e_{t,i} + a_{t,i} \tag{27}$$

Therefore, a node can be set to 0 if it is no longer being accessed. Meanwhile, a node can be updated through $a_{t,i}$ if it needs to be described in more than one word. In this way, we updated the node features from $X_t$ to $X_{t+1}$ to generate the next word.

## 4. Experiments

### 4.1. Experimental Datasets

An extensive set of experiments was performed on two widely used datasets: MSCOCO Entities [11] and Flickr30k Entities [41] to evaluate the effectiveness of the proposed model. Both datasets contained images with corresponding descriptions in English and a correspondence between nouns and image regions. The control signal of ASG was automatically constructed based on the annotations of two datasets, as in Ref. [12]. In order to have a fair comparison with other methods, we followed "Karpathy" splits for MSCOCO Entities, using 112,742 images for training, 4790 images for validation, and 4979 images for testing. Each image has almost five sentences, as well as a corresponding ASG control signal. As for the Flickr30k Entities, which are smaller than MSCOCO Entities, they have 29,000 images for training, 1014 images for validation, and 1000 images for testing. Each image has almost five sentences, and the corresponding ASG control signal does as well.

### 4.2. Experimental Evaluation Metrics

We evaluated the quality of the generated sentences through two aspects: accuracy and diversity. For accuracy, this paper employed six evaluation metrics, including BLEU@4(b@4) [42], METEOR (M) [43], ROUGE (R) [44], CIDEr (C) [45], SPICE (S) [46], and alignment score (NW) [11], where B@4 computes the precision of the generated words. However, BLEU@4 does not consider synonyms and part-of-speech information. The METEOR considers this information through WordNet and calculates the average of accuracy and recall. ROUGE is a similarity metric to computes the recall on the longest common subsequence. CIDEr assigns a lower weight to common words and a higher weight to novel words. This metric better reflects the matching level of novel words. Most of the novel words are objects, attributes, and relations, which would not be prepositions or adverbs. Hence, The CIDEr better reflects the matching level of the novel words. SPICE evaluates the semantic similarity of the generated sentences and ground truth. The alignment score (NW) evaluates the consistency between the generated caption and the regional sequence. For diversity, we followed Ref. [12] using two metrics: *n*-gram diversity (D-*n*) [13,47] and Self-CIDEr (s-C) [48]. The D-*n* is the ratio of the different *n*-grams to total number of words in the best five captions. The Self-Cider is a recent metric which uses the CIDEr score as the kernel matrix *K* in the LSA to evaluate semantic diversity. The range of BLEU@4, METEOR, ROUGE, SPICE, NW, D-*n*, and s-C is $[0, 1]$. The range of CIDEr is $[0, 10]$. Note that all the scores have been reported in percentages. The higher the score, the more accurate or diverse it is.

### 4.3. Experimental Details

This paper extracts visual features for grounded regions by standard Faster-RCNN [23] pretrained on VisualGenome, and we also extract global image features by ResNet152 [5] pertained on ImageNet. For the information-augmented graph encoder, the dimension of $C_1 = 2048, C_2 = 4096$, and $C_3 = 512$, using two layers of MR-WGCN. For the language decoder, the global feature dimension $C_g = 2048$, the word embedding and the hidden size of two LSTM were set to be 512. During training, we trained the network through

cross-entropy loss and the designed similarity loss over 25 epochs. For the Adam optimizer, the learning rate was set to 0.0002 and the batch size set to 128. For language decoding, we exploited the beam search strategy with a beam size of 5 for all experiments. All experiments were conducted on NVIDIA GPU GTX-1080Ti. IANR was based on ASG2Caption [12].

### 4.4. Ablation Experiment

To quantify the impact of each proposed module, we compared it with a list of ablation models on various settings. To ensure fairness, in the following experiments, we fix the initialization parameters of the network.

#### 4.4.1. Impact of Each Module on Encoder

To study the effects of the proposed information-augmented graph modules (memory-augmented attention (MA) [39], relational discriminator (RD), and multi-relational weighted graph encoder (MR-WGCN)) on the encoder, we started from a baseline model [12] that has a multi-relational graph encoder (MR-GCN) [40]. Then, we replaced MR-GCN in the baseline model with the proposed MR-WGCN, which takes into account how the message passes from one type of node to another along different edges. After that, we added MA and RD to the new model, respectively. Table 1 shows the results of each model on the two datasets.

**Table 1.** Settings and results of ablation studies. (Baseline: ASG2Caption [12]; MR-WGCN: replace MR-GCN in baseline model with the proposed MR-WGCN; MR-WGCN+MA: replace MR-GCN and insert memory-augmented attention(MA); MR-WGCN+MA+RD: replace MR-GCN with MR-WGCN, insert memory-augmented attention (MA) and relational discriminator (RD) into the baseline model). Bold for the best.

| Dataset | Model | B@4 | M | R | C | S | NW | D-1 | D-2 | s-C |
|---------|-------|-----|---|---|---|---|----|-----|-----|-----|
| MSCOCO Entities | baseline [12] | 23.06 | 24.63 | 50.31 | 204.36 | 42.39 | 70.46 | 47.63 | 73.51 | 69.95 |
| | MR-WGCN | 23.04 | 24.78 | 50.41 | 206.22 | 42.72 | 70.54 | 48.13 | 74.26 | 70.23 |
| | MR-WGCN+MA | 24.11 | 25.31 | 51.29 | 216.32 | 43.78 | 71.35 | 48.34 | 74.45 | **70.37** |
| | MR-WGCN+MA+RD | **24.39** | **25.48** | **51.54** | **218.12** | **44.16** | **71.62** | **48.51** | **74.66** | 70.33 |
| Flickr30k Entities | baseline [12] | 13.73 | 18.12 | 39.27 | 108.67 | 29.43 | 62.23 | 41.71 | 68.11 | 85.72 |
| | MR-WGCN | 13.83 | 18.45 | 39.57 | 111.58 | 29.45 | 62.92 | 40.33 | 66.06 | 85.08 |
| | MR-WGCN+MA | 13.85 | 18.36 | 39.52 | 112.71 | 29.74 | 62.75 | **42.37** | **69.06** | **86.82** |
| | MR-WGCN+MA+RD | **14.30** | **18.65** | **39.99** | **114.8** | **30.15** | **62.86** | 41.83 | 68.57 | 86.03 |

It is obvious that the model with MR-WGCN outperforms the baseline model on five types of accuracy metrics, except B@4. The score for B@4 on MSCOCO Entities is slightly lower than the baseline model by 0.02. In terms of diversity, the results on the MSCOCO Entities dataset are better than the baseline model, but worse on the Flickr30k Entities. This means the MR-WGCN can improve the accuracy of visual words significantly. The reason for MR-WGCN tending to generate similar sentences on small data sets may be the insufficient training data, which makes the MR-WGCN unable to learn the difference between each sentence. Then, we applied the MA module to add the prior information to the "MR-WGCN+MA" model. All evaluation metrics were improved. This means the MA module can steadily improve the accuracy and diversity of the generated sentences. The final model, which adopts the proposed RD module to evaluate the prior information, outperforms the "MR-WGCN+MA" model in all accuracy metrics. For the diversity, the results of the larger dataset are better than "MR-WGCN", but poor on the smaller one. Compared to the baseline model, better results are achieved for all metrics.

Figure 2 shows a few examples with images and captions generated by the ablated models with various settings and human-annotated ground truth sentences (GT). From these examples, the baseline model generates captions that are logical but inaccurate, while the proposed module generates more accurate captions. More specifically, the designed modules have advantages in the following three aspects: (1) IANR figures out the relationship between objects. There is a boat on the water with flags/motorcycle sitting

on a fencelaptop computer on the table in the first/second/third examples. However, the baseline model presents the flag as parked on a river/a bike sitting on a bridge/laptop computer in a room, while IANR describes it correctly; (2) IANR describes the objects in the control signals more accurately. For example, IANR describes the motorcycle and fence, not the bike and bridge, in the second example; the object is a laptop computer, not a desktop computer, in the third example; people are sitting in a park, not in a courtyard, in the fourth example; and (3) IANR counts objects more accurately. In the image of the five examples, there is one truck, not a couple of trucks. IANR has these advantages because it can add a priori information of objects and remove useless information.

| | |
|---|---|
| 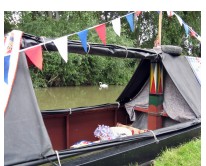 | GT: a boat that is decorated with flags on the water.<br>baseline: a boat and flag is parked on a river.<br>MR-WGCN: a boat in a lake that is next to a flag.<br>MR-WGCN+MA: a boat that is in front of a lake and a flag.<br>MR-WGCN+MA+RD: a boat that is on the water with flags. |
| 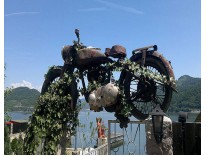 | GT: a motorcycle sitting on top of a fence as decor.<br>baseline: a bike sitting on a bridge in front of a field.<br>MR-WGCN: a bike sitting on top of a wall next to a tree.<br>MR-WGCN+MA: a motorcycle is parked on a fence in front of a field.<br>MR-WGCN+MA+RD: a motorcycle sitting on a fence in front of a boat. |
| 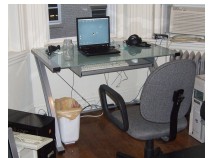 | GT: a desk that has a laptop computer on it.<br>baseline: a room that has a computer computer on it.<br>MR-WGCN: a room that has a laptop computer in it.<br>MR-WGCN+MA: a room that has a laptop computer on it.<br>MR-WGCN+MA+RD: a table that has a laptop computer on it. |
| 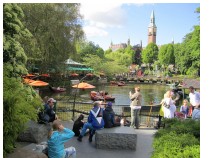 | GT: a group of people outdoors at a park.<br>baseline: a group of people are sitting outside in a courtyard.<br>MR-WGCN: a group of people are sitting on a path.<br>MR-WGCN+MA: a group of people are sitting on a bench.<br>MR-WGCN+MA+RD: a group of people are sitting in a park. |
| 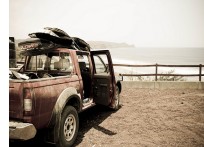 | GT: a red truck parked on top of a dirty ground.<br>baseline: a couple of trucks parked on a dirt field.<br>MR-WGCN: a very old truck parked on a dirt road.<br>MR-WGCN+MA: a large truck parked on top of a dirt field.<br>MR-WGCN+MA+RD: a red truck parked on top of a dirt field. |

**Figure 2.** Examples of captions generated by the baseline model, various ablation models mentioned in the Section 4.4.1, as well as the corresponding ground truths (GT).

### 4.4.2. Impact of Each Module on Decoder

To quantify the impact of each proposed module in the decoder, the ablation experiment is shown in Table 2. All models have MR-WGCN, MA, and RD modules in the encoder. The base model (Row 1 and 5) beginning with the decoder only has semantic content attention (SCA). Then, in Rows 2 and 6, we added a fusion constraint decode (FCD) to the decoder and the performance improved in the accuracy and diversity metrics, except for a b@4 drop of 0.01 in Flickr30k Entities. In particular, CIDEr/Spice/Self-CIDEr improved by 2.1/0.31/0.11 and 2.57/0.04/1.13, respectively. When comparing Row 2 with Row 3, in which a graph update (gupda) was employed for updating node features based on the generated words and currently accessed nodes, there was an improvement in all metrics except s-C. For the performance of Flickr30k Entities, the metrics of B@4, R, NW, s-C slightly decreased, while C, S, and M significantly increased. It shows the networks with gupda tended to generate similar prepositions when the training data were insufficient, and the increased M, R, C, S indicate that the nouns, relations, and so forth were described correctly. Rows 4 and 8 enhance the decoder with graph structure attention (GSA). The graph structure attention captures the structure information in the graph to supplement

semantic content attention. Hence, It outperforms other models in most metrics on two datasets. The metrics of METEOR/CIDEr/Spice/NW/D-1/D-2/Self-CIDEr improved by 0.1/0.98/0.27/0.03/0.52/0.51/2.54 and 0.04/0.42/2.78/0.7/0.63/0.55/0.63/0.11, respectively. The improvement on the small dataset was greater than that on a large dataset. The improvement for visual words was more obvious. Hence, the designed GSA is useful for generating more diverse and accurate sentences.

**Table 2.** Comparison of variants for proposed modules in decoder on MSCOCO Entities. (✓) indicates "used" (SCA:semantic content attention; GSA:graph structure attention; gupda:graph updating; FCD: Fusion Constraint Decode). Bold for the best.

| Dataset | # | SCA | GSA | Gupda | FCD | B@4 | M | R | C | S | NW | D-1 | D-2 | s-C |
|---------|---|-----|-----|-------|-----|-----|---|---|---|---|----|-----|-----|-----|
| MSCOCO Entities | 1 | ✓ | | | | 24.17 | 25.27 | 51.39 | 215.69 | 43.77 | 71.07 | 48.10 | 74.24 | 70.44 |
| | 2 | ✓ | | | ✓ | 24.35 | 25.45 | 51.52 | 217.79 | 44.08 | 71.16 | 48.38 | 74.53 | 70.55 |
| | 3 | ✓ | | ✓ | ✓ | **24.81** | 25.61 | **51.80** | 221.32 | 44.14 | 71.48 | 48.43 | 74.55 | 68.03 |
| | 4 | ✓ | ✓ | ✓ | ✓ | 24.67 | **25.71** | 51.73 | **222.3** | **44.41** | **71.51** | **48.95** | **75.06** | **70.57** |
| Flickr30k Entities | 5 | ✓ | | | | 14.11 | 18.29 | 39.70 | 110.15 | 29.72 | 61.90 | 41.64 | 68.33 | 86.12 |
| | 6 | ✓ | | | ✓ | 14.10 | 18.43 | 39.73 | 112.72 | 29.76 | 62.36 | 43.40 | 70.39 | 87.25 |
| | 7 | ✓ | | ✓ | ✓ | 13.98 | 18.59 | 39.70 | 114.50 | 29.99 | 62.31 | 43.47 | 70.59 | 87.10 |
| | 8 | ✓ | ✓ | ✓ | ✓ | **14.48** | **18.63** | **40.12** | **117.28** | **30.69** | **62.93** | **44.02** | **71.22** | 87.21 |

Figure 3 shows a few examples of images and the generated captions. From these examples, the proposed modules have advantages in the following two aspects: (1). IANR counts objects more accurately. There are two men/an elephant, not a young boy/some giraffes in the first and second examples. (2). IANR describes the objects more accurately. For example, an elephant, not some giraffes standing in the grass in the second example; a glass and a vase, not a glass and glass on a table in the third example; a teddy bear in front of a car dashboard, not on top of a black car in the fourth example. IANR has these advantages because it makes fuller use of the control signal to generate the correct sentences.

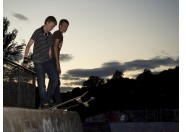
GT: two men on skateboards standing on the top of a ramp.
SCA: a young boy on a skateboard on the edge of a ramp.
SCA+FCD: two boys on skateboards riding on the edge of a ramp.
SCA+FCD+gupda: two boys on skateboards at the top of a ramp.
SCA+FCD+gupda+GSA: two men with skateboards standing at the top of a ramp.

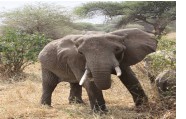
GT: an elephant standing in the grass on the plains.
SCA: some giraffes standing in a field of grass.
SCA+FCD:a zebra standing in the grass in a field.
SCA+FCD+gupda: an elephant standing in the grass in the grass.
SCA+FCD+gupda+GSA: an elephant standing in the grass in the sun.

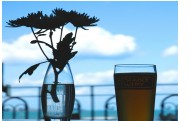
GT: a glass and vase sit on a table overlooking the ocean.
SCA: a glass and glass are on a table in front of a boat.
SCA+FCD: a glass and a glass are on a table in a city.
SCA+FCD+gupda: a glass and a vase are on a table with a drink.
SCA+FCD+gupda+GSA: a glass and a vase sit on a table near a pier.

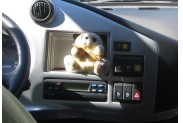
GT: the small stuffed bear is propped into the car dashboard.
SCA: a stuffed teddy bear sitting on top of a black car.
SCA+FCD: a stuffed teddy bear sitting on a computer dashboard.
SCA+FCD+gupda: a brown teddy bear sitting on top of a car dashboard.
SCA+FCD+gupda+GSA: a stuffed teddy bear sitting in front of a car dashboard.

**Figure 3.** Examples of captions generated by various ablation models. GT represents one of the corresponding ground truth sentences. SCA denotes the model has MR-WGCN, MA, and RD modules, but the decoder only has semantic content attention (SCA). SCA+FCD, SCA+FCD+gupda, and SCA+FCD+gupda+GSA is to add FCD, gupda, GSA modules into SCA step by step.

*4.5. Comparative Experiment*

Table 3 shows the performance comparison between the current state-of-the-art controllable and uncontrollable models with the proposed method. In this comparison, we use

the same MSCOCO Entities dataset set as Refs. [11,16,35]. Compared with other controllable image caption models, IANR was higher than SOAT in METEOR, CIDEr, and SPICE by 0.64, 50.58, and 4.91. All these enhancements show that IANR can significantly improve the ability to describe objects and relationships.

**Table 3.** Comparisons with the state-of-the-art on the MSCOCO Entities dataset. B@4, M, R, C, and S stand for BLUE@4, METEOR, ROUGE-L, CIDEr, and SPICE, respectively. Bold for the best.

| Model | B@4 | M | R | C | S |
|---|---|---|---|---|---|
| State-Of-The-Art Controllable Models | | | | | |
| MC [38] | **37.1** | 27.7 | 56.1 | 126.4 | 21.5 |
| ASG2Caption [12] | 23 | 24.5 | 50.1 | 204.2 | 42.1 |
| SCT [11] | 20.9 | 24.4 | 52.5 | 193 | 45.3 |
| VSR [16] | 23.1 | 28.0 | 55.6 | 235.1 | 48.9 |
| LCIC [15] | 35 | **27.9** | 57 | 116.8 | 21.7 |
| PC [32] | 36.4 | – | **57.5** | 124 | 21.2 |
| Sub-GC [36] | 36.2 | 27.7 | 56.6 | 115.3 | 20.7 |
| POS [13] | 31.1 | 25.3 | 53 | 103.6 | 18.8 |
| ConCap [49] | 40.5 | 30.9 | - | 133.7 | 23.8 |
| TSG [50] | 38.2 | 28.2 | 59.1 | 132.8 | 22.0 |
| ASA [51] | 44.0 | 32.0 | - | 140.4 | 23.8 |
| FVC-MT [35] | 22.4 | 25.8 | 55 | **206.3** | **47.6** |
| State-Of-The-Art Uncontrollable Models | | | | | |
| CPTR [52] | **40** | 29.1 | **59.4** | 129.4 | – |
| CGVRG [53] | 38.9 | 28.8 | 58.7 | 129.6 | 22.3 |
| AoANet [22] | 38.9 | **29.2** | 58.8 | **129.8** | **22.4** |
| BUTD [21] | 36.3 | 27.7 | 56.9 | 120.1 | 21.4 |
| SCA-CNN [54] | 30.2 | 24.4 | 52.4 | 91.2 | – |
| SCST [18] | 31.9 | 25.5 | 54.3 | 106.3 | – |
| IANR | 27.57 | 28.54 | 55.59 | **256.88** | **52.51** |

### 4.5.1. Comparison on the Same Test Data

A well-known advantage of controllable image captioning is the ability to generate diverse image captions through different control signals. Each control signal is produced in different ways, so some images or sentences will be removed. Hence, we compared IANR with the two latest controllable models, VSR [16] and SCT [11], which release codes, extract features, and pretrained models.

For fair comparison, those models were compared on the common parts of the VSR and SCT test datasets. The common part of the MSCOCO Entities has 4678 images and 14,179 sentences. The common part of the Flickr30k Entities has 1000 images and 4982 sentences. The input feature sequence of "SCT" is the region feature corresponding to the words in the sentence. The "SCT *w/o* sequence" generates sentences by predicting the sequence of the selected regions. "VSR" achieves a better score by specifying the ground verb and associated object node features. "VSR *w/o* verb" removes the ground verb information and only uses the features of the ground truth region and relations.

The quantitative results are shown in Table 4. It is obvious that the captions generated by IANR in two datasets have much higher accuracy and diversity (CIDEr 224 VS 165.66 in VSR, Self-CIDEr 50.05 VS 46.29 in VSR). "SCT *w/o* sequence" obtained the worst results because it lacked the ground region sequences. Compared with VSR, IANR does not only need to know what the relationship is. This is more consistent with the application of image captioning. Even though there is less information in control signals, the metrics of accuracy and diversity are still higher than VSR.

**Table 4.** The performance comparisons on MSCOCO Entities and Flickr30k Entities datasets. All tests were performed in the common part of the datasets, but each model had region features. Bold for the best.

| Dataset | Model | B@4 | M | R | C | S | NW | D-1 | D-2 | s-C |
|---------|-------|-----|---|---|---|---|----|-----|-----|-----|
| MSCOCO Entities | SCT | 18.27 | 26.20 | 50.44 | 151.62 | 40.27 | 71.57 | 48.99 | 67.26 | 39.85 |
| | SCT *w/o* sequence | 12.22 | 23.88 | 44.33 | 111.10 | 36.33 | 65.16 | 42.59 | 57.22 | 30.88 |
| | VSR | 16.00 | 28.91 | 49.24 | 165.66 | 38.62 | 65.67 | 52.34 | 70.93 | 46.29 |
| | VSR *w/o* verb | 14.98 | 27.91 | 47.73 | 146.95 | 38.60 | 65.15 | 51.64 | 70.20 | 45.28 |
| | IANR | **25.50** | **26.62** | **52.21** | **224.02** | **45.63** | **72.5** | **58.68** | **81.66** | **50.05** |
| Flickr30k Entities | SCT | 11.29 | 19.33 | 38.56 | 71.79 | 24.72 | 59.00 | 32.34 | 47.10 | 50.72 |
| | SCT *w/o* sequence | 9.79 | 17.99 | 35.81 | 60.57 | 23.18 | 56.54 | 29.60 | 42.62 | 42.76 |
| | VSR | 12.41 | 22.87 | 42.06 | 115.54 | 22.71 | 56.53 | 37.53 | 59.04 | 68.16 |
| | VSR *w/o* verb | 10.76 | 20.85 | 38.19 | 83.07 | 22.16 | 55.16 | 36.47 | 57.29 | 65.51 |
| | IANR | **14.60** | **19.01** | **40.58** | **116.98** | **31.27** | **63.53** | **48.14** | **74.40** | **73.72** |

### 4.5.2. Comparison on Same Training Data

We compared the latest model with the same test data in Section 4.5.1, but those models had different control signals and feature sequences. Hence, the proposed model is compared with several carefully designed baselines that use the same training data. Those baselines include: (1). AoANet, which employs self-attention as an encoder and decoder; (2). the BUTD model, which dynamically attends over relevant object regions when generating different words; (3). SCT, which regards the set of visual regions as a control signal; and (4) ASG2Caption, which proposes the ASG as a control signal.

Table 5 shows comparison results with the aforementioned models on MSCOCO Entities. IANR achieves state-of-the-art results on automatic evaluation metrics, outperforming all baselines in terms of alignment with the control signal through NW. IANR outperforms the controllable AoANet and controllable BUTD by 46.72 on CIDEr, 2.24 on NW, and 7.98 on Self-CIDEr. Compared with SCT trained with the same visual feature, our model improves by 93.48 on CIDER, 9.42 on NW and 17.17 on self-CIDEr. Finally, compared with the ASG2Caption model, IANR still outperforms it in all metrics, such as by being higher by 17.7 on CIDER, 0.91 on NW, and 0.62 on self-CIDEr.

**Table 5.** The performance comparisons on the MSCOCO Entities dataset for controllable image captioning. All models were re-implemented and trained on the same region feature. Bold for the best.

| Model | B@4 | M | R | C | S | NW | D-1 | D-2 | s-C |
|-------|-----|---|---|---|---|----|-----|-----|-----|
| AoANet [22] | 18.57 | 22.70 | 46.09 | 175.58 | 40.18 | 69.09 | 45.58 | 66.74 | 60.25 |
| BUTD [21] | 16.00 | 21.29 | 43.61 | 149.50 | 36.13 | 66.34 | 39.58 | 57.64 | 53.19 |
| SCT [11] | 14.29 | 22.51 | 44.49 | 128.82 | 34.99 | 63.61 | 40.80 | 58.41 | 51.06 |
| ASG2Caption [12] | 23.06 | 24.63 | 50.31 | 204.36 | 42.39 | 70.46 | 47.63 | 73.51 | 69.95 |
| IANR | **24.67** | **25.71** | **51.73** | **222.30** | **44.41** | **71.51** | **48.95** | **75.06** | **70.57** |

### 4.6. Result and Discussion

In the experiment, the results of some of the images were not correct. This section visualizes some failure results in Flickr30k Entities, as shown in Figure 4. On the left of the first line, there are too many boxes, resulting in captioning model failure to describe the relationships between them. On the right of the first line, according to the control signal, the model needs to describe populations and umbrellas that do not have relationships. Hence, IANR generates an incorrect description. The left side of the last two lines is the correct sentence and control signal, and the right side is incorrect. In the second line, it is difficult to describe the relationship between the child and the ocean. On the right side of the third row, the red box contains two boxes indicating a semicircle and the same direction, respectively, which is too close and makes the model difficult to describe.

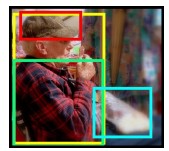 GT: an elderly man dressed in a red and blue plaid shirt and brown camouflage cap is watching an event while eating a snack from a plastic bag on his lap

IANR:an older man with a brown and white plaid shirt and a brown baseball cap is sitting on a stool with a blue hammer

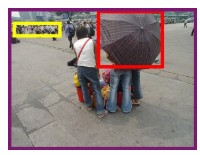 GT: three people stand underneath and umbrella in front of a crowd of people

IANR:a couple standing in the middle of a group of people are wearing an umbrella

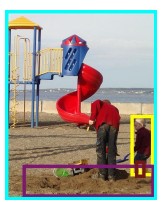 GT: two boys wearing red jackets are digging shovels into the dirt the smaller boy in the yellow boots has a Mohawk

IANR: a young blond haired child wearing a red sweatshirt and boots is playing on the dirt ground with a red toy in the background

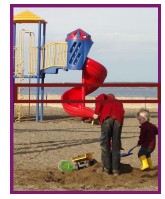 GT: two people dig with shovels at a playground by the ocean

IANR: a child rides a swing on a slide at the ocean

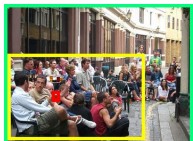 GT: some of the people in the crowd are having a drink

IANR: many people are sitting in a circle and drinking beer

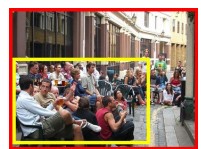 GT: people sitting around in a semi-circle all looking in the same direction

IANR: people sitting on a city street eating a busy street

**Figure 4.** Examples of ground truth and a failed case generated by the proposed model.

As can be seen from Figure 4, a proper control signal is the key issue. In the future, we propose to design an appropriate control signal and corresponding captioning model according to the actual application.

## 5. Conclusions

Consider that all currently available object-controllable image captioning methods have overlooked the prior information of detected objects and relationships. To this end, this paper proposed a novel module called the information-augmented graph encoder, which is composed of an information-augmented embedding module and a multi-relational weighted graph encoder. The dynamic attention model was designed to fuse the result of a control signal and node features with prior information. In addition, we designed a similarity loss for generating diverse captioning. Extensive experiments on the MSCOCO Entities and Flickr30k Entities achieved state-of-the-art performance in terms of controllable image captioning models. More remarkably, IANR exceeded the best-published CIDEr score to date by 6.7%/5.6% on the MSCOCO Entities/Flickr Entities test split. It also significantly improved the diversity of captions.

The main limitation of this study is the difficulty in constructing the control signal to determine what is needed to be described in the given image. However, in some specific applications, it is possible to know approximately what needs to be described, for example, describing the road or the surrounding items when assisting a visually impaired person to walk; and constructing a control signal through some models [55] to detect the salient object in remote sensing images. Compared to other image captioning models, IANR is able to run in real-time up to 1.15 ms per image on a GPU-enabled device, which is significantly faster. Hence, IANR is more suitable for combining with some detection models [55] to describe salient objects. In the future, it is proposed to simplify the control signals, compress the model, or combine it with some detection methods to make IANR available for mobile devices or real-time tasks.

**Author Contributions:** S.D.: conceived, designed the whole experiment, and wrote the original draft. Y.Z.: designed and performed the experiment. H.Z. (Hong Zhu): contributed to the review of this paper. J.S.: participated in the design of the experiments. D.W.: participated in the verification of the experimental results. N.X.: participated in the review and revision of the paper. G.L.: provided funding support. H.Z. (Huiyu Zhou): contributed to the review of this paper and provided funding support. All authors have read and agreed to the published version of the manuscript.

**Funding:** This research was supported by the NSFC No. 61771386, and by the Key Research and Development Program of Shaanxi No. 2020SF-359, and by the Research and development of manufacturing information system platform supporting product lifecycle management No. 2018GY-030, Doctoral Research Fund of Xi'an University of Technology, China under Grant Program No. 103-451119003, and by the Natural Science Foundation of Shaanxi Province No. 2021JQ-487, and by the Xi'an Science and Technology Foundation No. 2019217814GXRC014CG015-GXYD14.11, and by Natural Science Foundation of Shaanxi Province No. 2023-JC-YB-550.

**Institutional Review Board Statement:** Not applicable.

**Informed Consent Statement:** Not applicable.

**Data Availability Statement:** The data used to support the findings of this study are included within the paper.

**Conflicts of Interest:** The authors declare no conflict of interest.

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
