# Peer review of "Controllable Image Captioning with Feature Refinement and Multilayer Fusion"

_applsci, doi:10.3390/app13085020_

Round 1
Reviewer 1 Report
In this manuscript, a controllable image captioning model is presented based on a Multi-Relational Weighted Graph Convolutional Network (MR-WGCN). A similarity loss was utilized for model optimization. The authors proposed the information-augmented graph encoder, a module consisting of an information-augmented embedding module and a multi-relational weighted graph encoder. To demonstrate the effectiveness of the proposed approach, experiments and ablations were performed on the corresponding datasets (MSCOCO Entities, Flickr30k Entities) and performance was evaluated using the relevant metrics for accuracy and diversity (BLEU@4, METEOR, ROUGE, CIDEr, SPICE, NW, n-gram diversity, Self-CIDEr). The results are presented and discussed. Furthermore, the proposed approach is compared with other recent relevant approaches and the obtained results are discussed. The limitations of the presented approach are also listed. The authors have presented their work very well, both from a theoretical and practical point of view. The paper is technically sound and based on the fundamentals of the field, and the references provided by the authors are applicable and relevant (52 citations).
Please consider the following corrections and comments:
#) Regarding the sentence (Page 5, Line 179): “The effectiveness and accuracy of the region feature with prior information is improved by using our proposed relational discriminator structure. “The authors should support their claim with references and their own results.
Author Response
Thank you for your review. The proposed block is relational discriminator (RD). My claim is proved in Table 1. The metrics have significantly improved after adding the RD module.
Reviewer 2 Report
1. Can multiple captions possible for an object?
2. What are the issues or challenges in image captioning? some can be added in abstract
3. What evaluation metric is used? why?
4. fig 1 , what is input, what is output ? not clear
5. Author should also mention any wrong findings or wrong captions of proposed model
Reviewer 3 Report
The authors' work is well appreciated and it looks interesting in view of the readers perspective. The technical flow of the study is well presented.
The following are some observations .
1.Research articles published in 2022 should be referred and referenced to support the reported results
2.It is critical to remember that when communicating your research findings, the focus should be on the research rather than the people who conducted the research. When trying to persuade the reader, it is best to avoid using personal pronouns in academic writing, even if it is the authors' personal opinion.
3. It is suggested to improve the presentation style and Language
4. Suggested to include the proposed model's architecture
5.More detailed analysis description on the results reported in the tables
6. The more insight on the evaluation metrics , its range of values and its impact on the expected results need to be addressed.
Reviewer 4 Report
Overall, the article is good can be accepted with minimum improvement. Authors explained the research that they conducted and presented the result based on Figures and Tables shown in the article. In their work, latest journal was referred. But authors are required to:
1. Move Table 2 and Figure 3 in item 4.4.2 because its relate to the result related to this para.
2. One sub-topic required before Conclusion. The new sub-topic is Result and Discussion where authors are required to discusses about their findings and comments based on research conducted.
